# Circular RNA Profile in Atherosclerotic Disease: Regulation during ST-Elevated Myocardial Infarction

**DOI:** 10.3390/ijms25169014

**Published:** 2024-08-19

**Authors:** Fredric A. Holme, Camilla Huse, Xiang Yi Kong, Kaspar Broch, Lars Gullestad, Anne Kristine Anstensrud, Geir Ø. Andersen, Brage H. Amundsen, Ola Kleveland, Ana Quiles-Jimenez, Sverre Holm, Pål Aukrust, Ingrun Alseth, Bente Halvorsen, Tuva B. Dahl

**Affiliations:** 1Institute of Clinical Medicine, University of Oslo (UiO), 0372 Oslo, Norwayb.e.halvorsen@medisin.uio.no (B.H.); 2Research Institute for Internal Medicine, Oslo University Hospital, Rikshospitalet, 0372 Oslo, Norway; 3Department of Medicine, Cardiovascular Division, Brigham and Women’s Hospital, Harvard Medical School, Boston, MA 02115, USA; 4Department of Cardiology, Oslo University Hospital, Rikshospitalet, 0372 Oslo, Norwaykristine.anstensrud@gmail.com (A.K.A.); 5Department of Cardiology, Oslo University Hospital, Ullevål, 0450 Oslo, Norway; uxgend@ous-hf.no; 6Clinic of Cardiology, St. Olav’s Hospital, Trondheim University Hospital, 7030 Trondheim, Norway; 7Department of Circulation and Medical Imaging, Norwegian University of Science and Technology (NTNU), 7030 Trondheim, Norway; 8Department of Microbiology, Oslo University Hospital, Rikshospitalet, 0372 Oslo, Norway

**Keywords:** CircRNA, atherosclerosis, STEMI, tocilizumab

## Abstract

Circular (circ) RNAs are non-coding RNAs with important functions in the nervous system, cardiovascular system, and cancer. Their role in atherosclerosis and myocardial infarction (MI) remains poorly described. We aim to investigate the potential circRNAs in immune cells during atherogenesis and examine the most regulated during MI and the modulation by interleukin (IL)-6 receptor inhibition by tocilizumab. Wild-type (WT) and *ApoE^−/−^* mice were fed an atherogenic diet for 10 weeks, and the circRNA profile was analyzed by circRNA microarray. Whole blood from patients with ST-elevated MI (STEMI) and randomized to tocilizumab (*n* = 21) or placebo (*n* = 19) was collected at admission, 3–7 days, and at 6 months, in addition to samples from healthy controls (*n* = 13). Primers for human circRNA were designed, and circRNA levels were measured using RT-qPCR. mRNA regulation of predicted circRNA targets was investigated by RNA sequencing. The expression of 867 circRNAs differed between atherogenic and WT mice. In STEMI patients, circUBAC2 was significantly lower than in healthy controls. CircANKRD42 and circUBAC2 levels were inversely correlated with troponin T, and for circUBAC2, an inverse correlation was also seen with final infarct size at 6 months. The predicted mRNA targets for circUBAC2 and circANKRD42 were investigated and altered levels of transcripts involved in the regulation of inflammatory/immune cells, apoptosis, and mitochondrial function were found. Finally, tocilizumab induced an up-regulation of circANKRD42 and circUBAC2 3–7 days after percutaneous coronary intervention. CircRNA levels were dysregulated in STEMI, potentially influencing the immune system, apoptosis, and mitochondrial function.

## 1. Introduction

Atherosclerosis is the major underlying cause of cardiovascular disease (CVD), characterized by the formation of fibro-fatty, lipid-rich, and inflammatory lesions within the arterial wall. A crucial pathogenic factor in atherosclerosis is the bidirectional interaction between persistent low-grade inflammation and lipid abnormalities [1]. Treatment and prevention strategies for CVD have improved over the last 40 years [2]. Nevertheless, CVD remains one of the leading causes of morbidity and mortality worldwide [3]. Thus, there is still a need for the development of novel treatments and prevention strategies. An improved understanding of the pathogenic mechanisms of atherosclerosis and its clinical consequences, such as myocardial infarction (MI), is a prerequisite to achieve this goal.

Circular (circ) RNAs are non-coding RNAs in which the 3′ and the 5′ ends have been joined in a covalently closed loop through a process called back-splicing [4]. The lack of free terminal ends prevents the degradation of circRNA by exonucleases, making them more stable than their linear counterparts [5]. Levels of circRNAs vary at slow rates and are thought to be affected minimally by acute events [5]. These traits could make circRNAs potential biomarkers for chronic diseases, as suggested in cancer, Parkinson’s disease, and kidney disorders [6,7].

CircRNAs are shown to have regulatory roles for both transcription and translation [8]. The best-characterized mechanism in regulating transcription is by decreasing the levels of micro RNAs (miRNAs) by a process known as sponging. This binding of specific miRNAs to the circRNA reduces the amount of available miRNA and consequently increases the level of targeted messenger RNA (mRNA) [9]. Another suggested role is regulating their linear counterpart in cis, as back-splicing competes with linear splicing [10]. CircRNAs are suggested to regulate several pathophysiological processes with relevance to CVD, such as apoptosis [11], autophagy [12], inflammation [13], oxidative stress [14], and angiogenesis [15,16], potentially being therapeutic targets and biomarkers also in these disorders [16].

Several recent studies have investigated the effect of targeting inflammation in atherosclerosis and its clinical consequences, such as ST-elevated MI (STEMI) [17,18]. STEMI is one of the most serious outcomes of atherosclerotic disease and is associated with, and potentially also triggered by, enhanced inflammation and immune activation [19]. Several studies have shown a pathogenic role for interleukin (IL)-6-related pathways in STEMI by contributing to plaque inflammation, ischemia-reperfusion injury, and maladaptive myocardial remodeling [20,21,22]. Indeed, we have recently shown that treatment with the IL-6 receptor inhibitor tocilizumab in conjunction with percutaneous coronary intervention (PCI) improved the myocardial salvage index (MSI) in STEMI. This beneficial effect was associated with a downregulation of genes related to neutrophil function and, in particular, neutrophil degranulation [23]. If these tocilizumab-mediated effects also include modulation of circRNAs is still unknown.

In recent years, research has focused on describing the different circRNAs in human tissues and, in particular, their expression levels in different cancer types. However, although there are reports regarding circRNAs in atherosclerosis and MI, the profile of circRNAs in ischemic heart disease is still unclear. Additionally, the effect of anti-inflammatory therapy on circRNA levels in MI has so far not been investigated. To fill this knowledge gap, we first examined an atherogenic mouse model to discover circRNAs associated with atherogenesis. To examine the relevance of these findings to humans, we examined the expression of selected circRNAs in whole blood from patients with STEMI to examine (i) how circRNAs are regulated during and after STEMI; (ii) whether these molecules are related to outcome, i.e., MSI, troponin T (TnT) levels, and infarct size; (iii) the potential indirect effect of the regulated circRNAs on mRNA targets in the same patients; and finally, (iv) if and how levels of circRNAs are modulated by tocilizumab.

## 2. Results

### 2.1. Atherogenesis Alters the circRNA Expression: Pre-Clinical Explorative Studies

After 10 weeks on an atherogenic diet, *ApoE*^−/−^ mice (*n* = 3) developed atherosclerotic plaques in the aortic root, while no visible signs of atherosclerosis were seen in the WT mice (n = 3) (Figure 1A). Splenocytes, representing relevant immune cells in relation to atherosclerosis, analyzed for a total of 13.488 circRNAs, showed that a total of 867 circRNAs differed between atherogenic mice and WT mice (filtered by fold change ≥ 2 and *p* < 0.05). Levels of 596 circRNAs were significantly higher in *ApoE*^−/−^ mice than in the WT mice, and 271 circRNAs were significantly lower in the *ApoE*^−/−^ mice (Figure 1B). The 10 circRNAs with the largest enrichment in *ApoE*^−/−^ immune cells were more than 20-fold higher than in splenocytes from WT mice, and the 10 circRNAs with the largest suppression in *ApoE*^−/−^ splenocytes were 3.6–5.9-fold lower (Appendix A).

### 2.2. Atherogenic circRNA Translated to Human Homologs

To examine the relevance of these findings to humans, we next aimed to investigate the human analogs of the 10 most enriched and the 10 most suppressed circRNAs in the murine *ApoE*^−/−^ immune cells. Two had an intron origin and were excluded; five had already been identified, and the primer sequences were available (Figure 1C). For the remaining 13 circRNAs, primer pairs for the human homologs were designed with an assumption of similar back-splicing as in mice (Appendix A). Two primer pairs gave specific products only in macrophages (circKIF21A, circCACNB2), one only in whole blood (circRPPH1), and four both in macrophages and whole blood (circANKRD42, circEZH2, circTUBGCP3, circUBAC2) (Appendix A). Those that gave specific products in whole blood were used for further investigations in the STEMI patients (Figure 2).

### 2.3. circRNA Expression in Patients with STEMI

In the ASSAIL-MI trial, a single dose of intravenous tocilizumab was compared with placebo in patients with acute STEMI. The study drug was allocated in a 1:1 fashion. A total of 40 patients, 19 from the placebo arm and 21 from the tocilizumab arm, were chosen for circRNA analysis. The patients were selected based on age, sex, and clinical variables to obtain an equal distribution for the whole population.

Table 1 shows admission characteristics of the 40 patients who participated in this sub-study. At hospital admission, before administration of tocilizumab or placebo, circUBAC2 was significantly lower in the STEMI patients than in healthy controls (fold gene expression (FGE) 1.0 vs. 1.8; *p* = 0.001) (Figure 2). For circEZH2, circANKRD42, circRPPH1, and circTUBGCP3, however, there were no significant differences between STEMI patients at admission and healthy controls (Figure 2).

### 2.4. Negative Correlation between circRNA Levels and Peak TnT Levels and Final Infarct Size

In the whole study group, there were statistically significant correlations between peak TnT levels and circRNA levels at admission. As can be seen in Figure 3A, high levels of circANKRD42 and circUBAC2 were associated with lower levels of peak TnT (*r* = −0.530, *p* = 0.008 and *r* = −0.415, *p* = 0.009, respectively). A similar pattern was seen between the admission levels of circUBAC2 and the final infarction size at 6 months as assessed by CMR (*r* = −0.346, *p* = 0.033; Figure 3B), but not between circANKRD42 and final infarction size (*r* = 0.233, *p* = 0.284). In contrast, there were no significant correlations between these circRNAs and MSI (circANKRD42: *r* = 0.202, *p* = 0.344, and circUBAC2: *r* = 0.234, *p* = 0.151, respectively).

### 2.5. Regulation of mRNA Targets for the circRNA–miRNA Interaction

A proposed role for circRNAs is to sponge specific miRNAs and thus indirectly regulate gene expression in the cells [9,24]. According to published data, circUBAC2 [25] and circANKRD42 [26] are predicted to work as sponges for certain miRNAs with known functions relevant to the immune response in CVD [27].

The top 5 transcript targets for each miRNA were identified using the miRDB database [28,29] (Appendix A), and the transcript levels were investigated in the RNA-seq data from whole blood from STEMI patients [23]. Seven of the twenty-five investigated mRNA targets of circUBAC2-related miRNAs were differentially regulated, and all were lower at days 3–7 compared to the time at admission and tocilizumab treatment (Figure 4). Some of the most downregulated mRNAs targeted by circUBAC2 have been related to cell adhesion and inflammatory processes (*ERRFI1*) [30], regulation of apoptosis (*CREBRF)*, intracellular regulation of cholesterol metabolism (*AGFG1*) [31], and cell migration (*AFF4*) [32]. For the circANKRD42-related miRNA, nine of the twenty-five investigated targets were significantly regulated. Five of these were significantly higher and four were lower at days 3–7 vs. admission and tocilizumab treatment (Figure 4). Several of these targets have functions related to a proper immune response. *VDAC1* is known as a gatekeeper of mitochondrial function [33], while *SLAMF7* is related to B-cell and macrophage activation [34], *LETM2* is known to regulate the PI3K-AKT pathway [35], and *LACC1* is highly expressed in inflammatory macrophages [36]. Only one mRNA, *LETM2*, showed significant regulation between the placebo and tocilizumab-treated groups at 3–7 days (Log2FC −1.03, *p* = 0.0008).

Finally, another proposed role of circRNAs is to regulate their linear counterpart by competing with the linear splicing. However, there were no regulations of either linear *ANKRD42* or linear *UBAC2* in whole blood from STEMI patients at days 3–7 compared to admission.

### 2.6. circRNA Expression with IL-6R Inhibition

Levels of circANKRD42 increased significantly from admission to 3–7 days in the tocilizumab group (FGE 1.0 vs. 1.47; *p* < 0.05). In contrast, the modest increase in the placebo group did not reach statistical significance (FGE 1.0 vs. 1.26; *p* = 0.18) (Figure 5). circUBAC2 levels increased from admission to 3–7 days in the placebo arm (FGE 1.0 vs. 1.48; *p* < 0.05) as well as in the tocilizumab arm (1.0 vs. 1.91; *p* < 0.001). The increase was more pronounced in the tocilizumab group (FGE 1.91 vs. 1.48; *p* = 0.09) (Figure 5). Levels of circEZH2, circRPPH1, and circTUBGCP3 did not increase from admission in either the tocilizumab or the placebo group (Figure 5).

In the patients from the ASSAIL study that were selected for the circRNA analyses, there was a similar proportion of males and females. Notably, whereas circANKRD42 levels were slightly higher in females compared with males at 3–7 days in the placebo group, this difference reached statistical significance in the tocilizumab group (*p* = 0.012) (Appendix A).

## 3. Discussion

We show that the profile of circRNAs in splenocytes from atherosclerosis-prone *ApoE*^−/−^ mice treated with a high-fat diet differs from that observed in WT mice. To translate these findings to a human setting, we analyzed whether the human counterparts exist in whole blood from a STEMI population, randomized to receive IL-6 receptor inhibition by tocilizumab or placebo at hospital admission prior to/during PCI. We found that in STEMI patients, tocilizumab induced a marked up-regulation of two of the circRNAs that were highly differentially expressed between the *ApoE*^−/−^ mice and WT mice (circANKRD42 and circUBAC2). At the time of admission, the circUBAC2 level was lower in patients with STEMI than in healthy controls, suggesting that an up-regulation could be beneficial. It has previously been suggested that circRNAs are relatively stable and not influenced by acute events [5]. However, we show that certain circRNA are regulated during STEMI and modulated by anti-inflammatory therapy (i.e., tocilizumab).

Data from the preclinical study and the data from the clinical trial suggest some patterns of regulation of circRNAs. Levels of circUBAC2 were lower in the mice with atherosclerosis (*ApoE*^−/−^) than in the WT mice. Interestingly, patients with STEMI had significantly lower levels of circUBAC2 at admission compared to healthy controls, and intriguingly, tocilizumab induced a rise in circUBAC2 towards the levels observed in healthy controls. Whereas the levels of circUBAC2 rose in both treatment groups, the levels in the tocilizumab arm were closer to what we found in the healthy controls. All these results suggest that high levels of circUBAC2 could be beneficial. Additionally, admission levels of both circUBAC2 and circANKRD42 were inversely correlated with peak TnT, and for circUBAC2, a similar inverse correlation was seen with final infarct size at 6 months. It is therefore tempting to hypothesize that the more marked increase of these circRNAs in the tocilizumab as compared with the placebo group may be beneficial for the restoration of the myocardium and that levels of circUBAC2 could be used as a putative biomarker for myocardial inflammation and damage in the future.

In STEMI, circANKRD42 levels at admission did not differ from those of healthy controls. However, like circUBAC2, the levels of circANKRD42 rose during treatment with tocilizumab. Xu et al. suggest that circANKRD42 functions as a profibrotic factor in patients with idiopathic pulmonary fibrosis [26]. Although persistent pro-fibrotic activity may promote myocardial dysfunction, the rise in circANKRD42 soon after PCI could also contribute to tissue repair within the myocardium. Moreover, the predicted up-regulation of mRNA for *VDAC1* during tocilizumab treatment may improve mitochondrial function in MI [33]. In contrast to our results, Li et al. found up-regulation of circUBAC2 in whole blood from patients with MI. The two studies rely on different patient selections, i.e., only first-time MI in the ASSAIL-MI trial and only <6 h after symptom onset in the ASSAIL-MI trial. Nonetheless, there is a need for well-designed studies with strict protocols and validation cohorts to further elucidate the role of circRNAs in MI.

Previous research has shown that high levels of circRNA lead to more sponging of miRNA [9,24,37]. miRNA usually downregulates corresponding mRNAs [9]. In other words, high levels of circRNA lead to higher levels of mRNA due to less degradation. However, when comparing circRNA levels to mRNA levels in the ASSAIL-MI study, the results revealed that the 10 predicted miRNA targets only showed a total of five mRNA targets that were significantly higher, while 11 of the mRNA targets were significantly lower. Thirty-four of the investigated mRNA targets were not regulated. It is therefore tempting to hypothesize that circRNA sponging of miRNAs in these patients is deviating from what has been shown in vitro studies and that other roles of circRNA could be more relevant in vivo. Some of the predicted mRNAs that were regulated may be involved in the recruitment and activation of inflammatory cells (*ERRFI1*, *AFF4*, *SLAMF7*, *LACC1*) [30,32,34,36], apoptosis (*CREBRF*), intracellular regulation of cholesterol metabolism (*AGFG1*) [31], and mitochondrial function (*VDAC1*) [33]. All these processes are relevant in STEMI [38,39]. However, whether this regulation is due to or in coherent to the circUBAC2 and circANKRD42 levels needs to be further investigated. Regulation through circRNA protein binding is also a relevant pathway that needs attention [40]. Further studies are therefore needed to clarify the clinical relevance of the regulation of circRNAs in STEMI and the relevance of tocilizumab treatment.

Our study has some limitations. While we have an equal proportion of males and females in the STEMI population, we only conducted the pre-clinical trial on male mice, which potentially could have induced some bias in the selection of regulated circRNAs. The number of patients, and in particular healthy controls and mice, was relatively low, and we lack patients with chronic coronary atherosclerosis as a control group. Moreover, the pattern between male and female patients in the clinical part of the study should also be interpreted with caution due to the low number of patients in each group. The associations of regulated circRNA with mRNA targets are only predictive and should be interpreted with caution. The circRNAs in the pre-clinical study were examined in splenocytes that do not necessarily reflect the composition of immune cells as a whole, where neutrophils are dominating.

## 4. Materials and Methods

### 4.1. Ethics

The ASSessing the effect of Anti-IL-6 treatment in Myocardial Infarction trial (ASSAIL-MI) was approved by the Regional Ethics Committee (REK South-East 2016/1223) and the Norwegian Medicines Agency. All study participants provided written consent. The trial was conducted in compliance with the Declaration of Helsinki and Good Clinical Practice. The animal experiment was approved by the Norwegian National Animal Research Authority with FOTS project license numbers 7927, 8395, 5336, and 21681. All animal experiments were performed following the European Directive 2010/63/EU and conducted in accordance with the Animals in Research: Reporting in vivo Experiments (ARRIVE) guidelines.

### 4.2. Mouse Models

Male C57BL/6NTac (WT) (n = 3) and *ApoE*^−/−^ (n = 3) mice obtained from Taconic Biosciences (Cambridge City, IN, USA) were fed an atherogenic high-fat diet (40% fat, 0.27% cholesterol, Research Diets Inc., New Brunswick, NJ, USA, Diet #D16031603) ad libitum for 10 weeks before harvest. The atherosclerotic plaque burden was evaluated by histological analysis of the aortic root as described elsewhere [41]. Briefly, frozen hearts were embedded in the Tissue-Tek OCT compound and sectioned from the caudal to the cranial direction at 10 µm intervals on a cryostat. The presence of atherosclerosis was determined in paraformaldehyde-fixed sections collected at 100 µm intervals, stained with Oil-Red-O (Sigma-Aldrich, Darmstadt, Germany), and counterstained using hematoxylin (H-3404, Vector Laboratories, Newmak, CA, USA).

### 4.3. Murine Circular RNA Array

Total RNA was isolated from 20 million spleen immune cells from each WT and *ApoE*^−/−^ mice using the miRNeasy Mini Kit (Qiagen, Hilden, Germany). Isolated RNA was treated with DNase I (Qiagen) and stored at −80 °C until further analysis. RNA concentrations and purity based on the 260/280 and the 260/230 ratios were assessed by spectrophotometer absorbance (NanoDrop ND-1000, Thermo Fisher Scientific, Waltham, MA, USA). CircRNA microarray was performed using Arraystar Mouse circRNA Array V2 (Arraystar Inc., Rockville, MD, USA) based on Arraystar’s standard protocols. Briefly, RNase R (Epicentre, Inc., Lindenhurst, IL, USA) digested the total RNA to remove all linear RNAs to enrich the circRNAs. The enriched circRNAs were amplified and transcribed into fluorescent circRNA utilizing a random priming method (Arraystar Super RNA Labeling Kit; Arraystar, Rockville, MD, USA) and hybridized onto the Arraystar Mouse circRNA Array V2 (8 × 15K, Arraystar). After washing, the arrays were scanned by the Agilent Scanner G2505C. Agilent Feature Extraction software (version 11.0.1.1) was used to analyze the acquired array images. The R software (version 4.4), limma package, was used for quantile normalization and data processing. Differentially expressed circRNAs were visualized with Volcano Plot filtering. Differentially expressed circRNAs between two samples were identified through fold change (FC) filtering (FC > 5 and *p*-value < 0.05).

### 4.4. Primer Design

For human circRNA primers, we searched for human homologues based on the murine sequences found in the circRNA array. After comparing the sequences (Jalview, Oxford, GB, 2.11.3.2), we used the Primer-Blast tool: https://circinteractome.nia.nih.gov/circular_rna.html (accessed on 7 December 2022) to design the primer pairs. The primer pairs for circRNA analysis were also found in the literature: circUBAC2 [25], circANKRD42 [26], circEZH2 [42], circRPPH1 [43], circTUBGCP3 [44]. The specificities of the primers were assessed by gel electrophoresis and Sanger sequencing of the quantitative reverse transcription PCR (RT-qPCR) products (Eurofins Genomics, Ebersberg, Germany). The bands were extracted from the gel by using the QIAquick Gel Extraction Kit and protocol (Qiagen) (Appendix A).

### 4.5. Human Total RNA-Isolation and cDNA Synthesis

Total RNA was isolated from the BD PAXgene^TM^ Blood RNA tubes using MagMax^TM^ for Stabilized Blood Tubes RNA Isolation Kit (Invitrogen^TM^, Waltham, MA, USA) (ThermoFisher, Waltham, MA, USA) according to the manufacturer’s instructions. cDNA synthesis was carried out as described by Panda et al. [45]. Briefly, we mixed 650 ng total RNA, RiboLock (Thermo Scientific Lot, Waltham, MA, USA), dNTP mix (Thermo Scientific), random primers (Roche, Basel, Switzerland), 5x RT buffer (Thermo Scientific), and Maxima RT (Thermo Scientific). We incubated the samples for 10 min at 25 °C, followed by 30 min at 50 °C, and 5 min at 85 °C on a Veriti 96-well thermocycler (Applied Biosystems, Waltham, MA, USA).

### 4.6. Quantitative Reverse Transcriptase PCR

RT-qPCR was performed using the Bio-Rad CFX384 Maestro, C1000 Touch, qPCR machine (Bio-Rad Labratories Inc., Hercules, CA, USA) The optimal RT-qPCR protocol for our primer pairs was 3-step PCR, amplification + melt, 45 cycles with an annealing temperature of 56.6 °C. The RT-qPCR was performed on a CFX 384 real-time PCR system using 5 µL of cDNA and SYBR Green real-time master mix (Quantabio, Beverly, MA, USA). The expression levels of targeted circRNAs were normalized by the 2^−∆∆CT^ method. *Beta-ACTIN* and circHIPK3 mRNA levels were used as stable internal controls for the circRNA. The primers used in the PCR are listed in Appendix A.

### 4.7. miRNA and mRNA Targets

We predicted miRNA targets for circUBAC2 and circANKRD42 using miRanda [46], RNAhybrid [47], and TargetScan [48] algorithms. miRNA targets were also verified by the literature [25,26]. We investigated mRNA targets for miRNAs using the database miRDB (https://mirdb.org/mirdb/index.html, accessed on 30 January 2024) [28]. The top 5 mRNA targets for each miRNA were investigated in RNA sequencing from whole blood from patients in ASSAIL-MI.

### 4.8. RNA Sequencing

RNA sequencing from full blood was previously done and has been described in [23]. The sequencing was reanalyzed; briefly, the RNA samples were analyzed by Novogene (UK) Company Limited, Milton, Cambridge, GB. The ribosomal RNA depletion library preparation was used for the RNA isolated from whole blood with PAXgene tubes. We used the fastp software (v0.23.0) to remove contaminated adapters and low-quality reads with a phred score below 30 in the pair-end mode [49]. Filtered reads were mapped to the human transcriptome (Gencode Human Release H37, Heidelberg, Germany), and transcripts were quantified with 200 bootstrap iterations by Salmon (v1.5.2) [50,51]. We summarized the Salmon outputs to gene level and imported them into DESeq2 (v1.34.0) via tximeta (v.1.12.3) [52,53].

Descriptions of the mouse model, murine circular RNA array, primer design for human homologs, human RNA isolation, cDNA synthesis, quantitative reverse transcriptase PCR, prediction of miRNA and mRNA targets, and RNA sequencing are given in the Appendix A.

### 4.9. Patients and Study Design of ASSAIL-MI

In the phase 2 ASSAIL-MI trial (Clinicaltrials.gov: NCT03004703), a single dose of intravenous tocilizumab was compared with placebo in patients with acute STEMI. The study drugs were allocated in a 1:1 fashion to 199 patients between March 2017 and February 2020, as previously described [17]. The key inclusion criteria were STEMI and symptom onset less than 6 h before PCI. Exclusion criteria were previous MI; chronic infection, or chronic autoimmune or inflammatory disease; uncontrolled inflammatory bowel disease; ongoing infectious or immunologic disease; major surgery within the past eight weeks; or treatment with immunosuppressants other than low-dose steroids (equivalent to systemic exposure to 5 mg prednisone per day) [23]. A total of 40 patients, 19 from the placebo arm and 21 from the tocilizumab arm, were chosen out for circRNA and mRNA analysis. The patients were selected based on age, gender, and clinical parameters to obtain equal distribution as for the whole population. Admission characteristics of the 40 patients who participated in this sub-study are described in Table 1.

All the patients received dual antiplatelet therapy (DAPT) and unfractionated heparin (5000–7500 IE) intravenously before PCI was performed, and 76% of the patients had received unfractionated heparin (5000 IE) before arrival at the hospital.

### 4.10. Blood Sampling Protocol

We collected whole blood samples prior to the intra-arterial administration of unfractionated heparin and PCI. Blood samples were repeated 14–33 h after PCI and at 3–7 days, 3 months, and 6 months. We used BD PAXgene^TM^ Blood RNA tubes (BD, Franklin Lakes, NJ, USA) for RNA analysis of whole blood. For comparison of circRNA levels, we also collected venous blood samples from 13 sex and age-matched self-reported healthy controls.

### 4.11. Measurement of Cardiac Markers in the ASSAIL-MI Trial

High-sensitivity troponin T (TnT) was measured by electrochemiluminescence immunoassay (Elecsys 2010 analyzer, Roche Diagnostics, Basel, Switzerland). The myocardial salvage index (MSI) (%), defined as: $\frac{area\;at\;risk-infarct\;size}{area\;at\;risk}\cdot 100$ and final infarct size (% of left ventricular mass), was measured by cardiac magnetic resonance (CMR) imaging 3 to 7 days after the intervention as previously described [17].

### 4.12. Statistics

Statistical analyses were performed using SPSS version 25 (IBM Corp., Armonk, NY, USA) and GraphPad Prism 9.4.1 (GraphPad Software, Boston, MA, USA). Fold gene expression (FGE) was normalized toward the admission group, containing all patients admitted to the hospital, to get a fold change value of 1. An unpaired *t*-test was run to reveal significant differences in admission versus healthy controls. An unpaired *t*-test was also conducted for comparison of admission versus the different treatments (placebo vs. tocilizumab) and timepoints (admission vs. 3–7 days). A Pearson and Spearman correlation analysis was performed for admission circRNA levels and peak TnT, final infarct size, and MSI. For continuous variables, we used a two-sided *t*-test. Non-parametric variables were tested by using a Mann–Whitney test. We tested categorical variables using Fisher’s exact test. *p*-values < 0.05 (two-sided) were considered statistically significant.

## 5. Conclusions

To the best of our knowledge, this is the first study to investigate circRNA expression in immune cells from patients receiving anti-inflammatory treatment. We show that circRNA levels are altered in patients with STEMI. We also show an inverse correlation between investigated circRNAs and TnT and left ventricular mass. The circRNA levels are altered by tocilizumab. Our data suggest that the increase of circUBAC2 and circANKRD42 associated with treatment with tocilizumab can result in altered levels of mRNAs influencing the activation of the immune system, apoptosis, and mitochondrial function. However, further experiments and larger studies are needed to better understand the complex function of circRNAs in MI and how they could be modulated by therapy.

## Figures and Tables

**Figure 1 ijms-25-09014-f001:**
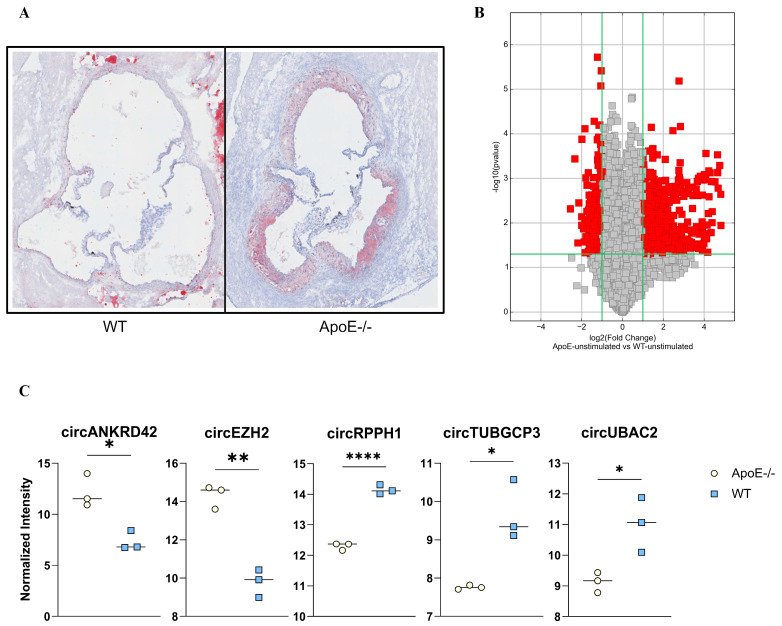
Murine circRNA profile in atherogenic *ApoE*^−/−^ mice. (**A**) Atherosclerotic lesions in the aortic root from wild type (WT) and *ApoE*^−/−^ after 10 weeks on an atherogenic diet, stained with Oil-Red-O. (**B**) Volcano-plot showing the circRNA regulation. Significantly differentially expressed circRNAs, with an FC > 5 and *p*-value < 0.05, from splenic cells in WT and *ApoE*^−/−^ mice marked in red. (**C**) Normalized intensity of the circRNAs from the circRNA murine array between *ApoE*^−/−^ vs. WT. * *p* < 0.05, ** *p* < 0.01, **** *p* < 0.0001.

**Figure 2 ijms-25-09014-f002:**
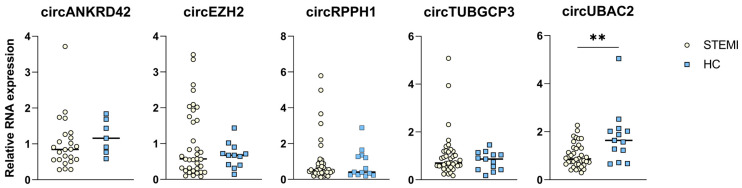
Relative circRNA expression of whole blood from STEMI patients at admission vs. healthy controls. The figure presents the FGE regulation of the circRNA targets in a subgroup of patients from the ASSAIL-MI. Samples were collected at admission before the patients were randomized to tocilizumab or placebo. Comparison between STEMI patients (*n* = 40) and healthy HC (*n* = 13) were made by an unpaired *t*-test. ** *p* < 0.01.

**Figure 3 ijms-25-09014-f003:**
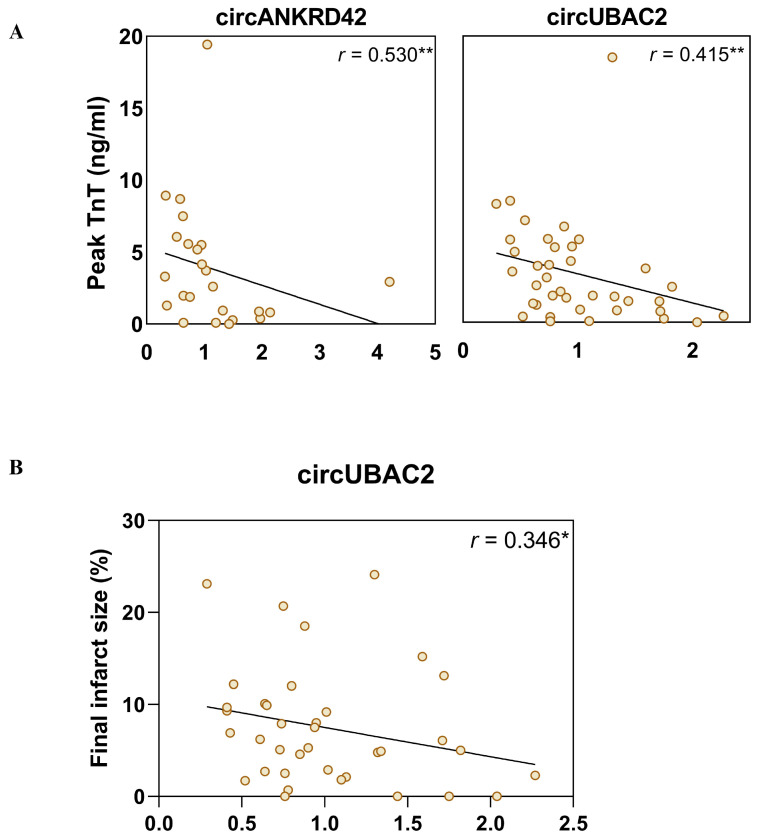
Correlation analysis between circRNA and clinical parameters from STEMI patients. (**A**) Inverse correlation with peak TnT and admission circRNA levels for circANKRD42 and circUBAC2 was conducted with a Spearman correlation analysis. (**B**) Inverse correlation between admission circRNA levels for circUBAC2 and the finale infarct size at 6 months, measured by % of left ventricular mass. * *p* < 0.05, ** *p* < 0.01.

**Figure 4 ijms-25-09014-f004:**
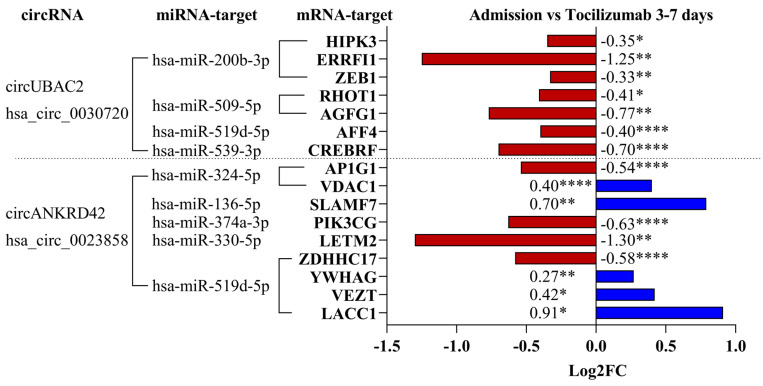
Overview of circRNA, miRNA, and mRNA interactions in patient samples. An overview of the miRNAs regulated by circUBAC2 and circANKRD42. We also show the mRNA targets for the different miRNAs and how they are regulated. * *p* < 0.05, ** *p* < 0.01, **** *p* < 0.0001.

**Figure 5 ijms-25-09014-f005:**
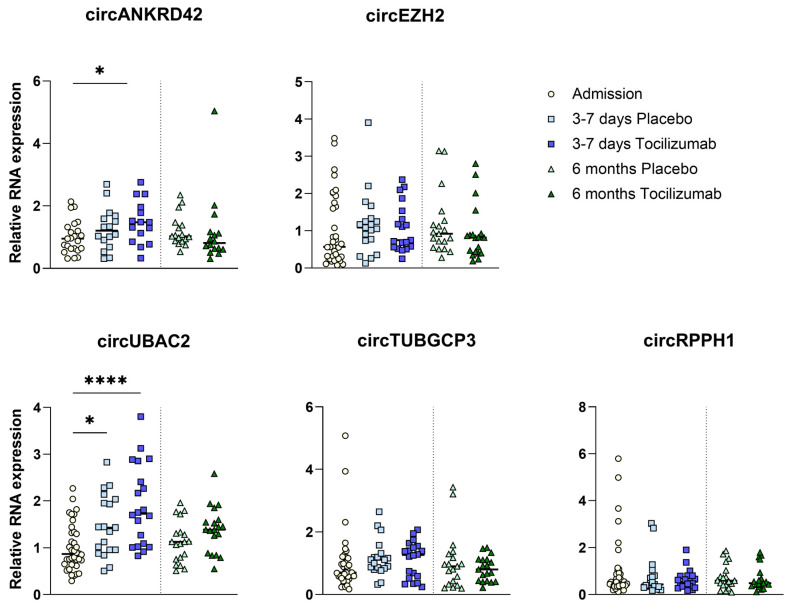
Fold gene expression of circRNA targets in immune cells in STEMI patients treated with tocilizumab (*n* = 21) or placebo (*n* = 19). The figure shows the regulation of the circRNAs after ST-elevation myocardial infarction (STEMI) at different time points in the tocilizumab and the placebo group. Unpaired *t*-test between the different groups. * *p* < 0.05, **** *p* < 0.0001.

**Table 1 ijms-25-09014-t001:** Overview of patient characteristics. Values are mean ± SD, n (%), or median (interquartile range). Admission characteristics stratified by treatment allocation and gender. ACE = angiotensin-converting enzyme; ARB = angiotensin receptor blocker; CK-MB = creatine kinase myocardial band; DAPT = dual anti-platelet therapy; HDL = high-density lipoprotein; LDL = low-density lipoprotein; PCI = percutaneous coronary intervention; NT-proBNP = N-terminal pro-B-type natriuretic peptide. The healthy controls (*n* = 13) were matched with age, sex, and ethnicity.

	Placebo (*n* = 19) A	Tocilizumab (*n* = 21) B
Demographics		
Age, years	64 ± 6	65 ± 6
Body mass index, kg/m^2^	24.4 ± 2.8	25.2 ± 3.3
Male	9 (47)	11 (52)
White	19 (100)	21 (100)
Smoking status		
Never smokers	6 (32)	7 (33)
Previous smokers	4 (21)	7 (33)
Current smokers	9 (47)	7 (33)
Prior conditions		
Angina pectoris	1 (5)	0
Cerebrovascular disease	1 (5)	1 (5)
Hypertension	7 (37)	7 (33)
Treatment		
ACE inhibitor or ARB	4 (21)	1 (5)
Oral anticoagulants	0	1 (5)
Platelet inhibitor	1 (5)	3 (14)
Beta-blocker	1 (5)	4 (19)
Calcium antagonist	2 (10)	3 (14)
Diuretic	1 (5)	0
Statin	3 (16)	5 (24)
Up-front DAPT	19 (100)	21 (100)
Clinical characteristics		
Blood pressure at admission, mm Hg		
Systolic	124 ± 22	130 ± 23
Diastolic	77 ± 14	77 ± 21
Heart rate at admission, beats/min	70 ± 16	70 ± 15
Time from symptom onset to arrival at PCI center, min	184 ± 80	144 ± 64
Door-to-balloon time, min	27 ± 15	21 ± 8
Killip class		
I	18 (95)	20 (95)
II	1 (5)	1 (5)
GRACE risk score	147 ± 20	147 ± 24
Infarct location		
Left anterior descending branch	3 (16)	5 (24)
Circumflex or marginal	5 (26)	2 (10)
Right coronary artery	11 (58)	13 (62)
Laboratory values		
Hemoglobin, g/dL	14.1 ± 1.3	13.9 ± 1.5
Platelet count, 10^9^/L	251 ± 52	261 ± 64
Total white blood cell count, 10^9^/L	10.5 ± 3.4	11.7 ± 4.0
Aspartate transaminase, U/L	29 (24–39)	29 (23–38)
Troponin T, ng/L	58 (28–95)	33 (19–62)
CK-MB, μg/L	6.0 (2.8–23.8)	4.4 (2.7–6.3)
NT-proBNP, ng/L	155 (58–347)	107 (52–212)
Creatinine, mmol/L	64 ± 13	70 ± 13
Glucose, mmol/L	8.1 ± 2.0	8.2 ± 1.3
HbA1c, mmol/mol	36 (33–48)	38 (36–39)
Total cholesterol, mmol/L	5.5 ± 1.1	5.4 ± 1.1
HDL cholesterol, mmol/L	1.3 (1.1–1.7)	1.2 (1.0–1.3)
LDL cholesterol, mmol/L	3.8 ± 0.9	3.9 ± 1.0
C-reactive protein, mg/L	4.1 (2.1–5.0)	1.7 (0.9–5.0)
Albumin, g/L	42 ± 6	40 ± 3

## Data Availability

Ethical restrictions from the Regional Committee for Medical and Research Ethics in South-East Norway, prohibits data from individual patients to be made available on publicly available repository. However, an institutional data transfer agreement can be established, and data can be shared if the aims of data use are covered by ethical approval and patient consent. The procedure will involve an update to the ethical approval as well as review by legal departments at both institutions, and the process will typically take 2 to 4 months from initial contact.

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
