# Peer review of "Circular RNA Profile in Atherosclerotic Disease: Regulation during ST-Elevated Myocardial Infarction"

_ijms, 2024, doi:10.3390/ijms25169014_

Round 1

Reviewer 1 Report

Comments and Suggestions for Authors

The paper „Circular RNA Profile in Atherosclerotic Disease – Modulatory Effects of Interleukin-6-Receptor Inhibition during ST-Elevated Myocardial Infarction” describes sub-study of ASSAIL-MI trial which aimed to investigate differences between circRNA in STEMI patients (on admission, 3-7 days after pPCI and after 6 months) comparing to healthy volunteers. The selection of study circRNA was conducted upon pilot experiment on mice (3 mice ApoE -/- and 3 control mice) with murine RNA microarray. Then, selected circRNA were referred to the human model (selection of primers, reverse transcription and cDNA elcetrophoresis and finally – Sanger DNA sequencing), expression levels were assessed by qRT-PCT in human individuals (as described above). Moreover, target miRNAs and mRNAs were predicted for selected circRNA and the expression of mRNA was assessed by RNA sequencing. In general, this study is well designated and adequate methods were used to investigate the matter. Nevertheless, there are two major flows that I my opinion needs correction:

1.      The hypothesis and aim of the study – the problem is what the Authors want to investigate. In this form of manuscript the aspect circular RNA profile and IL-6 pathway is mixed. It should be clearly divided, there are two options:

a.      Option 1 – concentrate on circRNA profile (murine microarray, selection of primers, expression measurment in healthy patients and STEMI patients, correlention between circRNA levels and infarct size/TnT, circRNA-miRNA-mRNA interaction and then – a subsection about circRNA expression under IL-6R blockade by tocilizumab (currently – subsection 2.4). (I’d suggest this option)

b.      Option 2 – concentrane only on IL-6R inhibition through the entire results section.

2.      The subsection 2.6 - Regulation of mRNA targets for the circRNA-miRNA interaction- Figure 5 describes the difference of mRNA expression (assessed through sequencing) in patients on admission and after 3-7 days of tocilizumab treatment. However, there are no information about other intergroup correlations. Especially, the presentation of differences in mRNA expression after 3-7 days on tocilizumab treatment vs placebo would be precious.

There are also some minor flaws, I’ll mark two of them:

1.      IL-6R blockade is a crucial point in this study, therefore more information on IL-6 pathway and its interference with RNA expression profile should be provided (not only brief information about the results of the previous study conducted by the Authors).

2.      This information is provided in „Methods” section, but I’d suggest that it should also be provided in the „Results” section, in every place when appropriate – „the circRNA plasma level in STEMI patients ….” Ecc.

Comments on the Quality of English Language

I'm not English native speaker, I have FCE certificate in English, according to me the quality of English is good, some minor corrections might be anticipated.  

Author Response

The paper „Circular RNA Profile in Atherosclerotic Disease – Modulatory Effects of Interleukin-6-Receptor Inhibition during ST-Elevated Myocardial Infarction” describes sub-study of ASSAIL-MI trial which aimed to investigate differences between circRNA in STEMI patients (on admission, 3-7 days after pPCI and after 6 months) comparing to healthy volunteers. The selection of study circRNA was conducted upon pilot experiment on mice (3 mice ApoE -/- and 3 control mice) with murine RNA microarray. Then, selected circRNA were referred to the human model (selection of primers, reverse transcription and cDNA elcetrophoresis and finally – Sanger DNA sequencing), expression levels were assessed by qRT-PCT in human individuals (as described above). Moreover, target miRNAs and mRNAs were predicted for selected circRNA and the expression of mRNA was assessed by RNA sequencing. In general, this study is well designated and adequate methods were used to investigate the matter. Nevertheless, there are two major flows that I my opinion needs correction:

Response. We thank the Reviewer for a thorough evaluation of the manuscript that also include valuable suggestions for improvement. We are also glad that the Reviewer found it of interest. 

Comment 1.      The hypothesis and aim of the study – the problem is what the Authors want to investigate. In this form of manuscript the aspect circular RNA profile and IL-6 pathway is mixed. It should be clearly divided, there are two options:

  1. Option 1 – concentrate on circRNA profile (murine microarray, selection of primers, expression measurment in healthy patients and STEMI patients, correlention between circRNA levels and infarct size/TnT, circRNA-miRNA-mRNA interaction and then – a subsection about circRNA expression under IL-6R blockade by tocilizumab (currently – subsection 2.4). (I’d suggest this option)
  2. Option 2 – concentrane only on IL-6R inhibition through the entire results section.

Response 1. We share this valid concern of the Reviewer and the Introduction is now less focused on IL-6 inhibition and in the Result, the former 2.4 (IL-6 receptor blockade by tocilizumab) is mowed to the end of the Results section. Accordingly, we have also modified the title of the manuscript. Thus, we clearly support the Reviewer´s option 1.  

Comment 2.      The subsection 2.6 - Regulation of mRNA targets for the circRNA-miRNA interaction- Figure 5 describes the difference of mRNA expression (assessed through sequencing) in patients on admission and after 3-7 days of tocilizumab treatment. However, there are no information about other intergroup correlations. Especially, the presentation of differences in mRNA expression after 3-7 days on tocilizumab treatment vs placebo would be precious.

Response 2. This is a valid comment and in the revised manuscript, we have included more information on the most downregulated mRNAs targeted by circUBAC2 with relevance to STEMI such as ERRF11, CREBRF and AGFG1 that relate to cell adhesion/inflammatory processes, regulation of apoptosis and cholesterol metabolism, respectively. Moreover, as suggested by the Reviewer, we have included data on the significant downregulation of LETM2 in the tocilizumab as compared with the placebo group at 3-7 with days. Although LETM2 is linked to PI3K-Akt Signaling Axis and apoptosis in cancer (Zhou S et al. A LETM2-Regulated PI3K-Akt Signaling Axis Reveals a Prognostic and therapeutic target in pancreatic cancer. Cancer 2022;14:472), the role of this newly discovered gene in MI is uncertain, maintaining nuclear integrity

Comment 3. 1. There are also some minor flaws, I’ll mark two of them:

  1. IL-6R blockade is a crucial point in this study, therefore more information on IL-6 pathway and its interference with RNA expression profile should be provided (not only brief information about the results of the previous study conducted by the Authors).

Response 3. 1 We have given more information on IL-6 pathways and their interference with STEMI in the revised manuscript (Introduction and Results).

  1. Comment 3. 2This information is provided in „Methods” section, but I’d suggest that it should also be provided in the „Results” section, in every place when appropriate – „the circRNA plasma level in STEMI patients ….” Ecc.

Response 3. 2 We have included more information on the ASSAIL study that was the basis for the clinical part of this manuscript in the Result section.

Reviewer 2 Report

Comments and Suggestions for Authors

I attach my comment in PDF file

Author Response

Comment: This manuscript highlights the importance of circRNA expression in CVD.

Because CVD is one of the leading causes of morbidity and mortality worldwide, I think it is very important to try identifying a novel parameter that might cause the disease. Furthermore, identifying the different expressions of circRNA in patients and healthy people can be considered as a promising diagnostic biomarker or a novel therapeutic. Thus, in my opinion this kind of paper is important. Considering this specific manuscript, I think the topic is important. Yet, before it is published, some minor revisions are required to improve the manuscript:

Response: We thank the Reviewer for a thorough evaluation of the manuscript that also include valuable suggestions for improvement. We are also glad that the Reviewer found it of interest.   

Comment 1. Point1: In this manuscript, the authors chose to focus on circRNA expression in male, I think the authors should explain their choice and if there is a different in the incidence of CVD between male and female.

Response 1. We agree that it could be a bias that circRNA in the pre-clinical model were only studied male mice, and in the revised, this is pointed out as a limitation of the study. Nonetheless, in the human study examining the regulation of circRNA during STEMI, there was a similar proportion of males and females. Notably, whereas circANKRD42 levels were slightly higher in females compared with males at 3-7 days in the placebo group, this difference reached statistical significance in the tocilizumab group (p=0.012). These results are included as a new Supplemental Figure 2. However, in these subgroup analyses the n was rather low. As pointed out in the revised manuscript, sex differences in the regulation of circRNA during MI needs to further studied in larger study populations to examine the clinical relevance of these findings.   

Comment 2. Point2: Line59-61: please check this sentence

“This binding of specific miRNAs to the circRNA reduces the amount of available miRNA and consequently the level of targeted messenger RNA (mRNA) [9]”

Because according to the known role of circRNA in sponging miRNA, it lead to an increase in the level of the mRNA target.

Response 2. We thank the Reviewer for this clarification of the language. We have now edited the sentence in the revised manuscript.  

Comment 3. Point 3: Table1: may the authors consider add information about the health people.

Response 3. We clearly apologize for not presenting information on healthy controls in the manuscript. We recruited 13 healthy controls based on disease history including the absent of any regular medications. The healthy controls were matched with the STEMI patients concerning age, sex and ethnicity. We have included this information in the legend to Table 1. We have also mentioned the rather low number of healthy controls as a limitation of the study.

Comment 4. Point 4: Line191: according to the data in figure 5, the two most downregulated mRNAs targeted by circUBAC2, are: ERRFI1and CREBRF. Maybe the authors can consider add more information about the role of the other mRNAs and how it can affect CVD, if there is a known role/ pathway, this can improve the manuscript.

Response 4. This is a valid comment and in the revised manuscript, we have included more information on the most downregulated mRNAs targeted by circUBAC2 with relevance to STEMI such as ERRF11, CREBRF and AGFG1 that relate to cell adhesion/inflammatory processes, regulation of apoptosis and cholesterol metabolism, respectively. Moreover, as suggested by the Reviewer, we have included data on the significant downregulation of LETM2 in the tocilizumab as compared with the placebo group at 3-7 with days. Although LETM2 is linked to PI3K-Akt Signaling Axis and apoptosis in cancer (Zhou S et al. A LETM2-Regulated PI3K-Akt Signaling Axis Reveals a Prognostic and therapeutic target in pancreatic cancer. Cancer 2022;14:472), the role of this newly discovered gene in MI is uncertain, maintaining nuclear integrity

Comment 5. Point 5: Line62-64: please add more references

Response 5. We have now included additional relevant references in the text.

Comment 6. Point 6: Line91-94: please check again the numbers, the total of 271circRNA that lower in ApoE-/- mice and 596 that higher in ApoE-/- mice is 867.

“analyzed for a total of 13.488 circRNAs, showed that a total of 876 circRNAs differed between atherogenic mice and WT mice (filtered by fold change ≥ 2 and p < 0.05) Levels of 596 circRNAs were significantly higher in ApoE-/- mice than in the WT mice, and 271 circRNAs were significantly lower in the ApoE-/- mice (Figure 1B).”

Response 6. We apologize for this mistake. As pointed out by the Reviewer, the total number of differently regulated genes are 867. We have corrected this throughout the manuscript. 

Comment 7. Point 7 : Figure 1c: Missing image reference in the text.

Response 7. Again, we apologize for this mistake that is corrected in the revised manuscript.